# From the Discovery of Targets to Delivery Systems: How to Decipher and Improve the Metallodrugs’ Actions at a Molecular Level

**DOI:** 10.3390/pharmaceutics15071997

**Published:** 2023-07-21

**Authors:** Ilaria Iacobucci, Sara La Manna, Irene Cipollone, Vittoria Monaco, Luisa Canè, Flora Cozzolino

**Affiliations:** 1UMR7042 CNRS-Unistra-UHA, Laboratoire d’Innovation Moléculaire et Applications (LIMA), European School of Chemistry, Polymers and Materials (ECPM), 67087 Strasbourg, France; 2Department of Chemical Sciences, University of Naples Federico II, 80126 Naples, Italy; cipollone@ceinge.unina.it (I.C.); monacovi@ceinge.unina.it (V.M.); flora.cozzolino@unina.it (F.C.); 3Department of Pharmacy, University of Naples Federico II, 80131 Naples, Italy; sara.lamanna@unina.it; 4CEINGE Biotecnologie Avanzate “Franco Salvatore” S.c.a r.l., 80131 Naples, Italy; canel@ceinge.unina.it; 5Department of Translational Medical Sciences, University of Naples “Federico II”, 80131 Naples, Italy

**Keywords:** metallodrugs, medicinal chemistry, chemoproteomics, thermal proteome profiling, pro-metallodrugs, delivery systems, nanoparticles

## Abstract

Metals are indispensable for the life of all organisms, and their dysregulation leads to various disorders due to the disruption of their homeostasis. Nowadays, various transition metals are used in pharmaceutical products as diagnostic and therapeutic agents because their electronic structure allows them to adjust the properties of molecules differently from organic molecules. Therefore, interest in the study of metal–drug complexes from different aspects has been aroused, and numerous approaches have been developed to characterize, activate, deliver, and clarify molecular mechanisms. The integration of these different approaches, ranging from chemoproteomics to nanoparticle systems and various activation strategies, enables the understanding of the cellular responses to metal drugs, which may form the basis for the development of new drugs and/or the modification of currently used drugs. The purpose of this review is to briefly summarize the recent advances in this field by describing the technological platforms and their potential applications for identifying protein targets for discovering the mechanisms of action of metallodrugs and improving their efficiency during delivery.

## 1. Introduction

The involvement of metals in many cellular and subcellular functions and their role in numerous vital processes is well known. Zinc (Zn), for example, is a mineral that is involved in numerous functions of cellular metabolism by supporting the catalytic activity of over 100 enzymes [1]. Calcium (Ca) is essential for cell physiology, and its transfer across membranes serves as a signal for many cellular processes, such as muscle contraction and the transmission of nerve impulses [2]. Copper (Cu) plays a key role in the central nervous system (CNS) [3] and in hematopoiesis [4], which is the process of blood cell formation. In addition, Cu and Cu-dependent enzymes are cofactors of numerous redox reactions and are involved, for example, in neurotransmission [5]. Evolution has made metals indispensable for the life of all organisms, and their dysregulation causes various, sometimes serious, disorders, such as a number of neurodegenerative diseases caused by the disturbance of their homeostasis [6,7]. It has long been known that metals can also serve as therapeutic agents, for example, as antimicrobial and antiviral weapons [8,9,10]. Recently, many metals have been incorporated into pharmaceutical products as diagnostic and therapeutic agents. The electronic structure of transition metals has the advantage of being very versatile in tuning molecular properties, unlike organic molecules [11]. The use of Pt coordination compounds in cancer therapy was certainly [12] one of the most unexpected developments in the field of medicine in the last 50 years. The first representative of chemotherapeutic agents, cisplatin, was discovered and synthesized by Michele Peyrone (1813–1883) [13], and its cytotoxic activity was revealed by chance by Barnett Rosemberg (1927–2009) in 1965. The clinical success of cisplatin as an anticancer drug accelerated the search for metals in medicinal chemistry [14]. Subsequently, several other Pt drugs, including carboplatin and oxaliplatin, were developed to improve the therapeutic efficacy and reduce the systemic toxicity caused by cisplatin [15,16,17].

Clinical studies have revealed controversial evidence. While some patients showed very positive results, others displayed the high toxicity of the compound [18]. The spectacular results obtained in patients suffering from tumors of the genital tract, particularly testicular cancer, in which cisplatin proved to be practically curative, led to its approval by the FDA (Food and Drug Administration) in 1978. Since then, cisplatin has been one of the most widely used drugs in cancer therapy worldwide and by far one of the most successful drugs (a so-called blockbuster drug) [19,20]. The spontaneous or induced resistance of tumors to cisplatin is an extremely serious problem, as it significantly limits the use of the drug. The main cellular processes by which cisplatin attacks tumor cells are as follows: uptake and transport, the formation of adducts with DNA and their recognition by specific proteins (damage-response proteins), the “translation” [21] of the DNA damage signal, which inhibits its replication and transcription and activates numerous signal transduction pathways that can lead to cell cycle arrest and damage repair or cell death via a variety of proteins that control cell growth, differentiation, and response to stress [22]. Research soon turned to other metals with the goal of finding complexes that would be effective against tumors resistant to platinum compounds and that might also have less systemic toxicity. For this reason, other metals have been tested as potential anticancer compounds, such as those containing ruthenium (Ru) [23], arsenic (As) [24], gold (Au) [25], and osmium (Os) [26], and they are currently being developed [14,27]. 

Among them, Ru compounds were the first to be studied, which is still a very active branch of research today, followed by arsenic (As) compounds, the only other non-radioactive metal (at the time) approved by the FDA (in 2000) for the treatment of tumors [28]. It was brought into clinical use in the 1970s for the treatment of numerous leukemias, especially acute promyelocytic leukemia [29]. At the moment, gold compounds are receiving increasing attention, as they have fewer tolerability limitations and appear to be target-specific [21]. Recent studies show that Au(I) complexes have antitumor activity, which is caused by inducing apoptosis in various cancer cell lines in vitro, comparing the effect of such compounds on resistant cells and analyzing the mechanism of action [30]. In addition, gold compounds have been proposed as potential anti-infective and anti-tuberculosis agents [31]. 

Despite the great potential of metal compounds, only a few have been finally accepted and marketed by the FDA because of toxicity and resistance induction, poor biodistribution, and short- and mid-term side effects [32]. Metallodrugs’ toxicity is mainly related to renal damage and neurotoxic effects [33], whereas resistance is due to targeting individual proteins or enzymes. On the other hand, the biodistribution and side effects could be improved by the further development of drug delivery strategies, such as nanoparticles [32,34]. The steps for the clinical approval established by the FDA are as follows: (1) discovery and development, (2) preclinical research, (3) clinical research, (4) FDA drug review, and (5) FDA post-market drug safety monitoring (https://www.fda.gov/patients/learn-about-drug-and-device-approvals/drug-development-process, accessed on 12 July 2023). All of these steps aim to determine the safety and efficacy of the drug under development in humans. Specifically, the clinical trial begins with Phase 0, a preliminary study in which some healthy volunteers are studied with an administration dose of less than 1% of the therapeutic dose for a maximum of seven days, followed by Phase I. After these phases, the metal drug can be tested in affected patients, which is Phase II of the clinical trials. The efficacy of the drug is tested in patients, and a placebo is used as a control. The final step is Phase III, which accounts for the majority of the clinical trials’ cost, and aims to confirm the safety and efficacy tested in Phase I and Phase II, respectively [35]. In Table 1, we report some of the metallodrugs that are currently under clinical trials, according to the FDA clinicaltrials.gov (accessed on 12 July 2023) portal. 

Nowadays, there are several metallodrugs approved by the United States and/or European Union (EU) countries for the medical treatment of human diseases. The application of these approved metallodrugs range from “Anticancer Metallodrugs” and “Therapeutic Radiopharmaceuticals or Phototherapeutic Metallodrugs” to “Antimicrobial and Antiparasitic Metallodrugs”, but also “Antidiabetes” and others (see ref. [35] for more information). Prior to the entire preclinical and clinical investigation, the first step is, of course, the investigation of one or more disease targets and the discovery of the molecular mechanisms of action of the metallodrug under investigation. In this review, we will discuss some of the key points regarding the investigation of metallodrugs as a therapeutic agents. In fact, the critical issues are the identification of the molecular targets of these drugs and the elucidation of their mechanisms of action, as well as their “delivery” and activation to the correct target. 

Functional proteomics aims to identify protein–protein (PPI) [36,37] and protein–DNA/RNA [38] interactions in vivo in order to define protein complexes and, thus, the cellular pathways involved in the biological processes of interest, as detailed in several papers and reviews [39,40]. There are many proteomic approaches used for the study of such interactions based on classical biochemical protocols adapted for the study of so-called “interactomics”, as well as coupling with advanced mass spectrometry instruments. This approach can also be used to define and identify the molecular partners of metal compounds in the context of chemical proteomics. In addition, drug delivery (i.e., the targeted administration of a compound to a tissue or cell where its controlled release ensures greater efficiency) is also an important area of study. Its application in the pharmacological field allows the transport of a molecule in our body, the selective delivery to the target tissue, and controlled release [41,42,43]. This method allows us to reduce the dose of the administered drug, reducing its possible side effects, and making it more bioavailable. Nanoparticles (NPs) currently represent a major advance in this field [32].

In this mini-review, we aim to provide an overview of metal-compound protein target purification and identification strategies from affinity purification approaches, including labeling and cross-linking (photo-affinity) strategies for the identification of transient interactions. In addition, label-free methodologies will be discussed. Furthermore, we aim to describe the different methods that are used nowadays to improve the efficiency of metallodrugs, in terms of activation and delivery in vivo, from the pro-drug approach to the different delivery systems. We believe that the study and combination of the two points (i.e., target identification and delivery) are milestones on the way to the clinical application of a metallodrug.

## 2. Protein Target Identification through Proteomic Approaches

The study of protein interactions with different classes of biomolecules has ancient roots in the field of proteomics, with many different biochemical strategies developed and advanced in conjunction with mass spectrometry techniques [39,40]. In recent years, several proteomic strategies coupled with mass spectrometry (MS) have emerged as a powerful and systematic approach for the large-scale identification of drug–protein interactions and the elucidation of their associated mechanisms.

Chemical proteomics, or chemoproteomics, is primarily devoted to the study of protein–small-molecule interactions and is attracting increasing attention in drug target discovery [44,45]. Chemoproteomics is a leading tool in the field of drug discovery that relies on affinity-based or label-free approaches to identify protein targets.

### 2.1. Affinity-Based Strategies

The affinity-based approach to chemical proteomics aims to immobilize the drug on a solid support and isolate the target proteins. This strategy is also referred to as drug pulldown [46,47]. Pulldown strategies and, more generally, affinity purification-mass spectrometry approaches (AP-MS), are proteomic techniques that allow us to isolate multiprotein complexes starting from a known molecule to hypothesize the intracellular processes in which it is involved [48,49,50].

Drug pulldown requires the following four main steps: (1) the immobilization of the pharmacophore on resin beads with affinity chromatography; (2) the isolation of the target molecules; (3) the identification of the target molecules with liquid chromatography–tandem mass spectrometry (LC–MS/MS); and (4) bioinformatics analysis to identify and quantify the proteins (Figure 1).

The advantage of pulldown techniques is that the physiological state of the proteins, the concentration levels (i.e., abundance), the post-translational modifications, and the natural binding partners are preserved by simply using non-denaturing lysis buffers. In particular, for Step 1 of the drug-pulldown workflow (Figure 1), different methods can be used to immobilize the active pharmacophore, which also depend on the stability of the organic- and inorganic-metal complexes. For this reason, Babak et al. used a biotin/streptavidin approach to immobilize a Ru(II) compound that is classified as a member of the RAPTA family [51]. The arene ligand of the complex was functionalized with a primary amine that can be biotinylated via an aminocaproic acid linker. The research results led to the identification of 15 cancer-related proteins that can explain the activity of RAPTA molecules, including *MK*, *PTN*, and *FGF3* as metastasis-related effectors, but also proteins related to cell cycle regulation, i.e., *GNL3*, *CGBP1*, *FAM32A*, and *VIR*. Drug biotinylation can also be performed by using the click reaction (Cu(I)-catalyzed alkyne–azide cycloaddition (CuAAC)). CuAAC is a reaction between an alkyne and an azide group, catalyzed by Cu(I) ions [52,53]. This reaction was used by Neuditschko and colleagues to derivatize the drug with biotin and fish the targets with streptavidin beads. Another clever method for the immobilization of metal drugs was reported by the same [54] research group. A complex was formed between the Ru(III) compound (i.e., BOLD-100) and human serum albumin (HSA). Then, the authors used the anti-HSA beads utilized for the human serum depletion to form the BOLD-100-HSA-beads adduct used to isolate drug targets [55]. Because high false positive rates occur in pulldown experiments, the authors also performed a competitive assay. The cell lysate was pre-treated with free BOLD-100 before exposure to the immobilized drug on the beads. This latter strategy allowed the saturation of the selective binding sites, as shown in Figure 2, so that the resulting target profile contains only non-selective binding partners. The subtraction of these from the target profile obtained by normal pulldown deletes the non-selective interactors and provides a list of selective binding partners. The competitive pulldown experiment is an alternative way to validate and remove the possible false-positive targets from a canonical pulldown experiment. Using this approach, the authors identified ribosomal proteins and the transcription factor *GTF2I* as BOLD-100 targets. In particular, further transcriptome analysis validated *RPL10* and/or *RPL24* among the ribosomal protein BOLD-100 targets.

Although the simple pulldown strategy is a straightforward approach, it suffers from the classic drawbacks that are generally associated with affinity purification strategies, such as the lack of transient interactions and the problem of false positives, as mentioned above (see [38] for specific further insights on this topic). For these reasons, the drug immobilization step is the most critical, as the final yield of target purification depends on it. In classical pulldown experiments with small organic molecules, the drug is usually covalently functionalized on an N-hydroxysuccinimidyl (NHS) sepharose carrier, then the non-functionalized portion of the carrier is blocked using several washes with ethanolamine [45]. For metal-based drugs, these types of washes are not compatible, because ethanolamine can compete with the metal complex ligands. Therefore, alternative methods have been developed, for example, using the biotin–streptavidin interaction (as described previously). The metal complex site carrying the label must be carefully selected in order to avoid interference with the biological activity of the pharmacophore. The CuAAC strategy is one of the best known for metal complex biotinylation and offers the advantage of high reaction efficiency and a favorable reaction time under mild conditions. However, a key drawback is the toxicity of copper ions to proteins. They can damage the structure and function of the target proteins and promote the formation of ROS [56]. In addition to the clickable part, i.e., the alkyne, a photo affine group can often also be produced [57]. Indeed, this strategy can be used to stabilize transient interactions, and it is also possible to stabilize the interactions in living cells, i.e., photoaffinity labeling (PAL). The photoactivatable group, usually a diazirine or benzophenone group, has the function of facilitating covalent adducts with the metal/metal-drug-interacting proteins upon light irradiation, as reported by Liu et al. [58]. The authors synthesized a photoactivatable molecule that enables the probe to form covalent adducts with the proteins interacting with the metal drug upon irradiation with UV light [59]. The authors incubated HeLa cells with a Au(III) meso-tetraphenylporphyrin (gold-1 a) compound and irradiated them at 365 nm to activate the benzophenone reaction with proteins via a radical mechanism. After cell lysis, an alkyne group on the probe was used to functionalize the protein–drug complexes with biotin in order to enrich the adducts on streptavidin beads. In their study, they identified heat shock protein 60 (Hsp60) as a target of gold-1a. These approaches can be used to identify interactions characterized by non-covalent and coordinative bonds. The covalent detection of drug–target interactions in cells can be used to identify physiologically occurring and moderately strong binding events by affinity-based chemoproteomics in vivo [60,61].

### 2.2. Label-Free Approaches

On the other hand, label-free approaches exploit protein stability due to the drug interaction.

For example, thermal proteome profiling (TPP) [62] has been used to study metallodrug targets [63]. TPP measures the extent of the drug–target interaction by monitoring the effects of pharmacological treatment on protein denaturation/solubility as a function of a progressive increase in temperature.

The extent of protein stabilization by the small molecule is proportional to the strength of the interaction and can be assessed by multiplexed mass spectrometry analysis (see [64] for a comprehensive description of all TPP applications). As shown in Figure 3, TPP can be performed by varying the temperature range (TR-TPP) (Figure 3A) or drug concentration (CR-TPP) (Figure 3B). After protein digestion and mass spectrometry analysis using LC–MS/MS, proteins are relatively quantified by the comparison between the drug-treated and drug-free conditions. The soluble fraction for each protein is reported as a function of the temperature or drug concentration.

TR-TPP was recently applied to discover the antitumor Bis(*N*-Heterocyclic Carbene)Pt(II) complex targets in intact cells, leading to the identification of asparagine synthetase (*ASNS*) [65]. Another label-free method used in the study of metallodrug targets is the functional identification of target by expression proteomic (FITExP) approach [66]. The basic principle is that protein targets and major mechanistically related proteins are modulated upon prolonged drug exposure; in particular, they are overexpressed when a lethal concentration (LC_50_) is administered to the cells. Lee et al. [67] applied the FITExP method to evaluate the mechanism of action of Ru(II) complexes, RAPTA-T, and RAPTA-EA. Although the approach suggested many protein targets, the best results were obtained in terms of revealing the mechanism of action of the molecules. For example, RAPTA-EA and RAPTA-T were found to induce the overexpression of several oxidative-stress-related and tumor-suppressing proteins, respectively [67]. FITExP analysis of proteins extracted from cells treated with RAPTA-T revealed, among the proteins identified, PLD3 protein, which belongs to the phospholipase D enzyme (PLD) family, with the lowest *p*-value. Instead, cells treated with RAPTA-EA had heat shock protein 1A/1B (*HSPA 1A/1B*) as the protein with the top-ranked *p*-value. This approach allowed the authors to hypothesize the following two different mechanisms of action for the two drugs: RAPTA-T targets are involved in multiple processes, suggesting a broader mechanism of action of RAPTA-T. In contrast, the targets of RAPTA-EA are exclusively involved in the regulation of the oxidative stress response. The combination of TPP and FITExP was performed in the work of Saei and colleagues to reduce the number of false positives and false negatives [63]. In principle, the other label-free strategies used in the study of protein–small-molecule interactions could also be applied to the field of metallodrugs. These methods encompass the protein stability based on the oxidation rate (SPROX), [68] pulse proteolysis (PP) [69], drug affinity responsive target stability (DARTS) [70], and limited proteolysis-coupled mass spectrometry (LiP-MS) [71]. However, to our knowledge, no work has been published.

Label-free techniques are capable of detecting the transient interactions of drug candidates with potential protein targets and globally assessing drug-altered temperature and the proteolytic or chemical stability of proteins, and otherwise do not require synthetic modification of the drug [72,73].

## 3. Methods to Enhance Metallodrugs Efficiency Administration

Another crucial aspect in medicinal chemistry is the development of innovative delivery systems that are able to recognize the target site or strategies to cause their selective activation in the target area. Typically, once introduced into a biological environment (serum, cell, or biomimetic environment), metal complexes can interact with biological ligands such as lipids, the amino acid residues of proteins, nucleic acids (DNA), or small molecules (e.g., vitamins, ions, neurotransmitters). This plethora of interactions can lead to severe, harmful side effects in most patients, limiting their use [74]. Recent advances in this field, such as the development of pro-metallodrugs or drug delivery systems (e.g., liposomes or solid-lipid nanoparticles), offer the opportunity to improve both the safety and the efficacy of metallodrugs and dispel the widespread myth of inherent toxicity. The first strategy, i.e., the preparation of prodrugs, aims to improve both the physical features of the metallodrug and its specificity toward pathological cells, while leaving normal cells unaffected. More specifically, in implementing the physical properties, recent developments in this field aim to achieve the following: (1) to shield the metallodrug complex and improve its biological properties, such as solubility, lipophilicity, redox stability, and cellular uptake, in order to ensure that it can be taken orally; and (2) to use the anticancer metallodrug complex as a carrier for an additional drug, thereby increasing the overall potency and efficacy of the drug, with the potential for a dual, or even multiple, mode of action [75]. Also, very interesting is the idea that pro-metallodrugs could discriminate between pathological and normal cells in cancer therapy, based on tumor-specific characteristics (e.g., oxygen concentration [76] and pH [77]). The problem of drug resistance has also been studied by inhibiting overexpressed enzymes in the tumor microenvironment [78,79]. As with the prodrug strategy, the development of delivery systems aims to increase the pharmacological efficiency of metallodrugs, while minimizing their side effects and implementing cell-target specificity. Some metallodrugs may also exhibit time-dependent or dose-dependent activity. Delivery systems can provide controlled-release mechanisms, allowing for sustained drug release over an extended period or a triggered release at specific sites. Interestingly, nanoparticle-based drug delivery systems have been shown to play a role in overcoming cancer-related drug resistance. Drug resistance is the tolerance of cancer cells to anticancer drugs and is a well-known phenomenon responsible for the limited success of chemotherapies. One example is cisplatin, the resistance to which is a major clinical obstacle [80]. Although the mechanism of cisplatin resistance is not fully understood, research efforts have focused mainly on reducing its deactivation, increasing its intracellular accumulation, and reducing DNA repair [81]. The advent of nanomedicines has brought breakthroughs in this field [82]. One of the best-known examples is NC-6004 (Nanoplatin), a cisplatin nanoparticle that was developed using state-of-the-art micelle nanotechnology. Cisplatin is encapsulated in polymeric micelles (30 nm in size) by forming a polymer–metal complex between polyethylene glycol-poly (glutamic acid) block copolymers (PEG-P(Glu)). NC-6004 uses the phenomenon known as the enhanced permeability and retention (EPR) effect to selectively penetrate and accumulate in tumor lesions. Unlike conventional cisplatin, NC-6004 can circulate in the bloodstream for a prolonged period of time. This prolonged circulation is enabled by the outer PEG envelope that protects the micelle from being retained by the reticuloendothelial systems. As a result, NC-6004 gradually releases cisplatin through the exchange of chloride ions in the human body [83]. Figure 4 provides an overview of the activation/delivery strategies discussed below.

### 3.1. Pro-Metallodrugs

Unlike most conventional small molecule drugs, metal complexes are often “prodrugs” that are activated en route to or at the target site (Figure 4, Pathway 1). There are many different activation strategies that have been investigated, and we have presented some recent examples here, as follows:
(1)Activation via hydrolysis

Hydrolysis is a common activation mechanism for transition metal drugs, in which weakly bound σ-donor ligands are displaced by water (A in Figure 4). Square-planar Pt(II) complexes are certainly the most commonly used. The breakthrough compound in this family is undoubtedly cisplatin, namely cis-[Pt^II^Cl_2_(NH_3_)_2_], a well-known anticancer drug used to treat a variety of cancers. Cisplatin activates as soon as it enters the cell, and its mode of action begins within the cell with the hydrolysis of Pt–Cl bonds to form a Pt–H_2_O complex, from which more reactive mono-aquatized [Pt^II^(OH_2_)Cl(NH_3_)_2_]^+^ and/or di-aquatized [Pt^II^(NH_3_)_2_(OH_2_)_2_]^2+^ species are formed [50]. These intracellular products can react with DNA, where they cause cell cycle arrest and apoptosis. Unfortunately, the problem with cisplatin is that it can be inactivated to transplatin, trans-[Pt^II^Cl_2_(NH_3_)_2_], during uptake into the cell. To avoid this problem, several other derivatives of cisplatin, with a similar mechanism of action (MOA), have been synthesized, where two ligands are linked together and the transaction is reduced. One of the best-known examples is carboplatin, [Pt(NH_3_)_2_(CBDCA-O,O′)], where CBDCA is a cyclobutane-1,1-dicarboxylate. Similar to cisplatin, half-sandwich pseudooctahedral Ru(II) and Os(II) η6-arene-diamine complexes, [Ru^II^/Os^II^(η6-arene)(N,N)Cl]^+^, also hydrolyze and bind to DNA, but monofunctionally, only one labile monodentate ligand [84].

(2)Redox activation

Altering the redox balance is an effective anticancer strategy because of the marked redox susceptibility of cancer cells, including hypoxia. The vast majority of reported examples consist of Pt(IV) complexes, which are less active than Pt(II) analogs and are reduced at the tumor site because of the higher levels of glutathione (GSH) and other antioxidants present there [53]. The general MOA consists of the following three main steps: the octahedral Pt(IV) complexes reach the cancer cells intact, they are then activated by reductive elimination of the axial ligands (the two accepted electrons enter the dz2 orbital (LUMO), which destabilizes the ligands in the axial positions), leading to the final release of the cytotoxic Pt(II) complex, thus acting as a prodrug (B in Figure 4) [54]. Four octahedral Pt(IV) prodrugs have entered clinical trials, namely, tetraplatin, iproplatin, satraplatin, and LA-12; however, none of them are currently approved for clinical use. As axial ligands, they may contain bioactive carboxylate molecules, as follows: (i) specific molecules targeting the tumor, such as steroids, folates, amino acids, and peptides; (ii) enzyme inhibitors; and (iii) anticancer drugs that have targets other than DNA and may synergize with the action of the cytotoxic Pt(II) metabolite. Recently, Ravera and co-workers investigated several cisplatin-based Pt(IV) complexes with dual action containing, as axial ligands, anticancer drugs such as ketoprofen (2-(3-benzoylphenyl)propanoic acid) or naproxen (2-(6-methoxynaphthalen-2-yl) propanoic acid). These complexes were found to act synergistically. The presence of non-steroidal anti-inflammatory drugs (NSAIDs) in the structure increases the lipophilicity of the complex and facilitates its cellular accumulation. Once inside the tumor cells (HCT 116 and A-549), cisplatin is released after Pt(IV) → Pt(II) reduction and, together with the NSAID, induces the activation of NAG-1, a protein that has antitumor and pro-apoptotic effects [53]. Similarly, a recent study developed a Pt(IV) combination containing, as an axial ligand, an active antimetastatic metabolite of limonene, 4-isopropenylcyclohexene-1-carboxylic acid, or perillic acid (PA). Also, in this case, the increase in lipophilicity resulted in enhanced cellular accumulation and the consequent release of the cisplatin component and PA-ligand-induced cytotoxic and antimetastatic effects, respectively [85].

Interestingly, Pt(IV) complexes are being studied not only as anticancer agents, but also as inhibitors of amyloid aggregation. It has been widely reported that Pt(II) complexes can modulate Aβ peptide aggregation by coordinating amino acid side chains [58]; however, the main problem with these complexes is their poor uptake by the brain, which limits their use in vivo. Kenche and co-workers investigated a novel Pt(IV) complex [Pt^IV^(*N*,*N*-dimethyl-2-[2-(quinolin-8-yl)-*1H*-benzimidazol-1-yl] ethanamine) Cl_4_]. The Pt(IV) complex showed an increased uptake in the brain compared with the Pt(II) complex, and may limit peptide aggregation and toxicity in cortical neurons after reduction to Pt(II). The treatment of an APP/PS1 mouse model of Alzheimer’s disease (AD) showed a statistically significant reduction in CSF Aβ_1–42_ levels and a reduction in plaque load [86].

(3)Photoactivation (Light-activatable metallodrugs)

Metallodrugs can be selectively activated with high spatial resolution in cancer cells in photodynamic therapy (PDT), photothermal therapy (PTT), and photoactivated chemotherapy (PACT).

Ru(II) complexes are a class of molecules known for their rich photochemistry. They can undergo photoinduced ligand dissociation, and the resulting Ru(II) aqua species can covalently bind to DNA in manner similar to cisplatin [87] (C in Figure 4. This process is associated with ^1^MLCT (metal-to-ligand charge transfer) transitions. Upon irradiation with appropriate light, Ru(II) complexes first reach the ^1^MLCT state, and then the ^3^MLCT state, through ultra-fast intersystem transition. The excited ^3^MLCT state of Ru(II) complexes can return to the ground state through non-radiative inactivation or luminescence pathways, or interact with other molecules such as O_2_ to generate singlet oxygen (through intersystem crossing to intra-ligand (IL, ILCT) states), showing their potential as photodynamic agents. They can also populate the ^3^MC state (metal-centered state or ligand-field) by thermal activation, which can lead to ligand dissociation and to the formation of Ru(II) aqua species with DNA-binding ability, showing their potential for photoactivated chemotherapy [74].

Ru(II)-TLD1433 is a novel water-soluble photosensitizer currently in clinical trials. It has unique properties, including selectivity for bladder tumors. In the dark, it has low cytotoxicity, whereas after activation with green laser light, it generates cytotoxic singlet oxygen (^1^O_2_) and radical oxygen species (ROS), which cause cancer cell death [88].

As an active agent for PDT, another metal complex that does not contain Ru(II) has been approved in the EU since November 2017: TOOKAD^®^ soluble (Pd II). It is a palladium bacteriopheophorbide monolysotaurine, also known as WST11, that is used to treat adenocarcinomas of the prostate. It is a derivative of bacteriochlorophyll, the photosynthetic pigment of certain aquatic bacteria that derive their energy from sunlight, and becomes pharmaceutically active when it is illuminated with light. TOOKAD is retained in the vascular system and, upon its activation using laser light with a wavelength of 753 nm, generates oxygen radicals that cause local hypoxia, which induces the release of nitric oxide (•NO) radicals. This leads to transient arterial vasodilation that then triggers the release of the vasoconstrictor endothelin-1. The rapid consumption of the •NO radicals by oxygen radicals leads to the formation of reactive nitrogen species (e.g., RNS and peroxynitrite), in parallel with arterial constriction [89,90].

### 3.2. Delivery Systems

As such, the administration of metallodrugs can often be problematic, due to their rapid metabolism, difficulty in reaching the site of action, and high systemic toxicity. To overcome these limitations, research has turned to finding novel delivery systems that are capable of adequately transporting and protecting the drug. The most studied delivery systems are generally polymeric or inorganic nanoparticles (NPs) and liposomes (Figure 4, Pathway 2).

Studies in this area have focused mainly on Pt chemotherapeutics to overcome the drawbacks associated with the use of this class of drugs in clinical cancer chemotherapy. A liposomal formulation of cisplatin, called Lipoplatin, reached Phase III of clinical trials. Lipoplatin, which is a nanoparticle with an average diameter of 110 nm, composed of lipids and cisplatin, can evade immune surveillance so that it is not eliminated by macrophages and can extravasate into a tumor through the damaged endothelium of the vascular system [91,92].

Nanoparticle systems have also been developed for two Ru(III)-drugs, KP1019 and NAMI-A, which have reached clinical trials. A nanoscale drug conjugate of (NAMI-A)-block copolymer micelles showed an improved inhibition of cell invasion and migration while enhancing the antimetastatic activity compared to the metallodrug alone in pancreatic and ovarian cancer cells [93,94]. The encapsulation of KP1019 in poly(lactic acid) (PLA) nanoparticles containing Tween-80 resulted in higher cytotoxicity than KP1019 alone in hepatoma cell lines and colon carcinomas [95].

A recent study reported the design of diruthenium(II,III)-NSAID metallodrugs encapsulated in biocompatible terpolymer–lipid nanoparticles (TPLNs) to target glioblastoma cancer. The metal complex was formed with a Ru_2_(II,III) mixed-valence metal–metal multiply bonded core linked to four carboxylate ibuprofen (Ibp) drug ligands, [Ru_2_(Ibp)_4_Cl]. Its encapsulation in TPLNs significantly enhanced the antiproliferative effect in two human glioblastoma cancer cells, U87MG and T98G, which are chemoresistant to cisplatin [96].

Since a perfect drug delivery system should be characterized by high biocompatibility, stability, and selectivity for a specific target site, research is moving towards the use of delivery systems composed of molecules that are already present in our bodies, such as ferritins (Fts). These are natural proteins that are involved in the storage and release of iron, can self-assemble into hollow, cage-like structures, and are recognized by receptors that are overexpressed on the surfaces of cancer cells. These proteins were recently selected to encapsulate a prototype of a new class of metallodrug with a PtAs(OH)_2_ core, called Arsenoplatin-1(AP-1). Cellular experiments on human epidermoid carcinoma cell lines and human keratinocytes (A431 and HaCaT) showed significantly higher selectivity of AP-1-loaded Ft against cancer cells compared to normal cells [1].

## 4. Conclusions and Future Directions

The use of drugs containing metal centers is suggested for the treatment of various classes of pathologies. The purpose of this short review is to highlight how these techniques can contribute to research in this important scientific area in order to advance ambitious hypotheses about the introduction of new anticancer drugs in their clinical development and use. There are numerous reviews illustrating the various chemical proteomic techniques that allow the study of metal–protein interactions in a biological system. Of course, much depends on what type of investigation is to be performed, what type of information is to be obtained, and what kinds of interactions are to be preserved. Presently, increasingly advanced techniques of affinity purification, photolabeling, and quantitation combined with mass spectrometry techniques allow us to easily identify putative interactors of metal–drug complexes, preserving labile interactions and providing useful information on the drug mechanisms of action. Therefore, it remains important to study the behavior of different metal complexes in order to overcome resistance problems and potentially reduce toxicity. The information obtained from chemical proteomic analysis could also be useful in understanding the mechanisms by which metal-based drugs reach their targets.

In this area, significant progress has been made in the development of prodrugs and delivery systems. The use of metallodrugs as prodrugs allows for the targeted and controlled activation of therapeutic effects, minimizing off-target effects and increasing efficacy. The incorporation of metal ions into delivery systems results in improved drug stability, controlled release, and targeted delivery to the desired site, which increases the therapeutic potential [70].

The future of metallodrugs seems very promising. Continued research and technological advances will be essential for optimizing prodrug design, delivery systems, and targeting strategies, which may lead to the improved specificity, efficacy, and safety of metallodrug-based therapies. Although much progress has been made in the field of metallodrugs, much more needs to be improved in order to fully exploit the potential ability to tailor them to specific diseases and patient populations, paving the way for personalized medicine. In addition, the integration of metallodrugs with emerging fields such as nanotechnology, bioconjugation, and regenerative medicine opens new avenues for innovative therapeutic approaches. Ultimately, advances in metallodrugs—from prodrugs to delivery systems—represent the first step in the development of next-generation metal-based therapeutics with improved clinical outcomes.

## Figures and Tables

**Figure 1 pharmaceutics-15-01997-f001:**
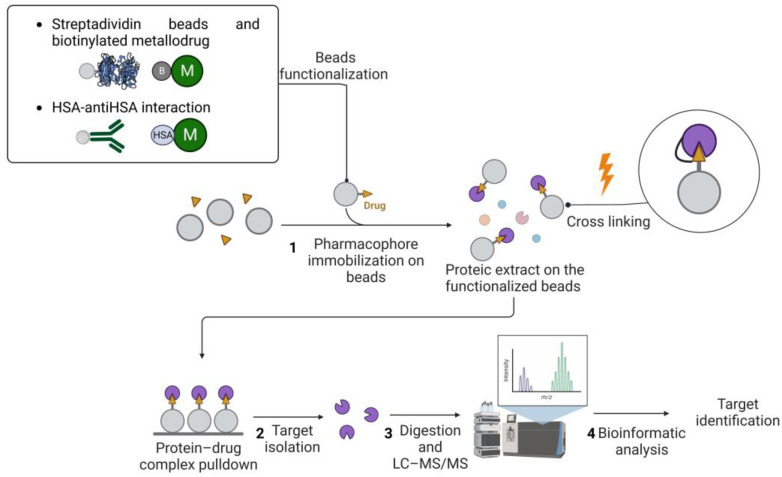
Drug pulldown consists of four steps: (1) immobilization of the pharmacophore on beads, which can be performed using different approaches (streptavidin beads and biotinylated metallodrugs or human serum albumin (HSA)–antiHSA antibody interaction); (2) target isolation, which can be preceded by a stabilization step involving UV-crosslinking between the proteic target and the drug; (3) mass spectrometry analysis of the target; and (4) bioinformatic analysis for the targets identification. Legend: M: metallodrug; B: biotin. Created with BioRender.com.

**Figure 2 pharmaceutics-15-01997-f002:**
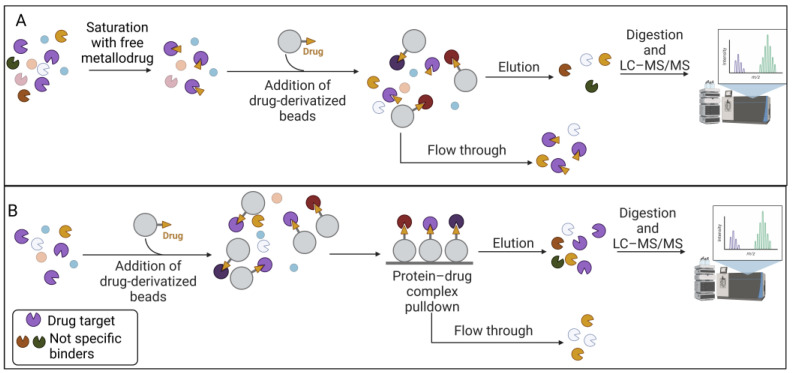
Competitive pulldown workflow. (**A**) The total protein extract was incubated with the free metallodrug and then subjected to the chemoproteomic strategy. Only the non-specific binders were eluted. Instead, the specific target was not retained on the beads. (**B**) The total protein extract was incubated with the metallodrug-derivatized beads for the chemoproteomic strategy. The non-specific and specific proteins were eluted. The comparison between the eluted proteins in the competitive and non-competitive will highlight the false-positive content. Created with BioRender.com.

**Figure 3 pharmaceutics-15-01997-f003:**
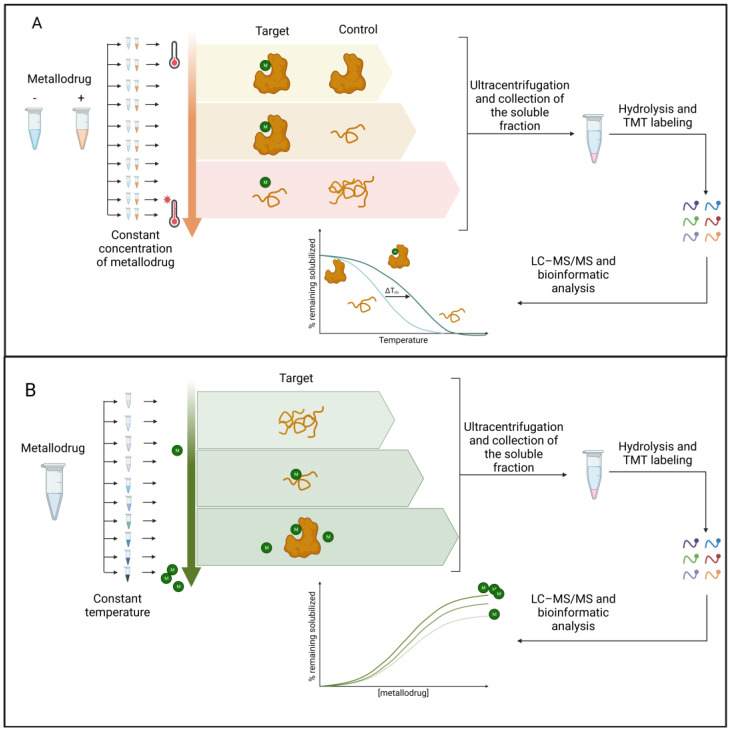
(**A**) TR-TPP experiment. The protein extract is incubated with both the vehicle and the metallodrug. Ten aliquots are prepared for each condition. Each aliquot is heated at a specific temperature, and then the soluble fractions are digested with trypsin and labeled with a TMT isotope tag. The samples are analyzed with LC–MS/MS and the protein is identified. Melting curves are fitted and melting temperatures are calculated for all proteins. (**B**) CR-TPP experiment. Untreated protein extract, as a control, and those treated with nine different concentrations of metallodrug, are heated to the same temperature. The soluble fractions are then digested with trypsin and labeled with a TMT isotope tag. The samples are analyzed using LC–MS/MS and proteins are identified and quantified. Dose–response curves are fitted, and thermal stability parameters are calculated. Legend: M: metallodrug. Created with BioRender.com.

**Figure 4 pharmaceutics-15-01997-f004:**
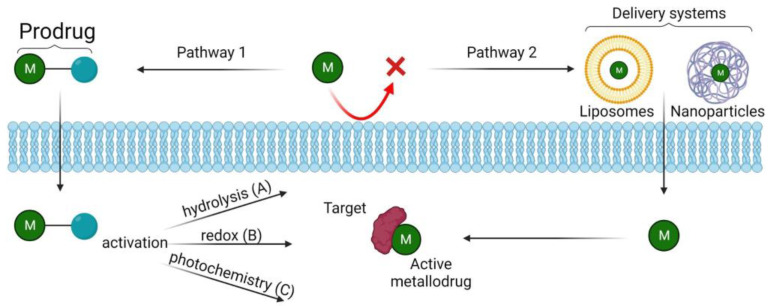
The most common strategies to improve the bioavailability of metal drugs: Pathway (1), pro-metallodrugs: the metallodrug is bound to a probe that allows the molecule to pass through the phospholipid bilayer. Then, the metallodrug can be activated by hydrolysis (A), redox activation (B), or photochemical activation (C); Pathway (2), the metallodrug is transported through carrier systems (e.g., liposomes, nanoparticles) that properly transport and protect the drug. Created with BioRender.com.

**Table 1 pharmaceutics-15-01997-t001:** Some of the metallodrugs currently under clinical investigation according to clinicaltrials.gov (accessed on 12 July 2023) portal. For each drug, we report the name and/or description; the mechanism of action; the application and, in brackets, the clinical trial ID number; and the phase of the study. N.A. not annotated.

Name/Description	Mechanism of Action	Application (ID)	Clinical Trial Phase
Picoplatin	DNA damage, cell division arrest, and apoptosis	Treatment of: Solid tumor (NCT00710697); Bladder cancer Breast cancer Colorectal cancer (NCT00465725); Lymphoma Small intestine cancer Unspecified adult solid tumor, protocol specific (NCT00016172)	Phase I Phase II Phase I
Satraplatin	DNA-adduct alteration and translational DNA synthesis inhibition	Treatment of: Prostate cancer (NCT01289067, NCT00634647, NCT00069745); Lung cancer (NCT00370383, NCT00268970); Breast cancer (NCT00265655); Brain cancer (NCT01259479); Advanced cancers (NCT00473720)	Phase II/II/III Phase II/II Phase II Phase I, Phase I
Lipoplatin/(Liposomal form of cisplatin)	DNA damage, ERK pathway, and apoptosis	Treatment of: Pleural effusion, Malignant (NCT02702700)	Phase I
Triplatin tetranitrate/BBR3464/CT-3610	DNA damage and apoptosis	Treatment of: Pancreatic cancer (NCT00024362) Lung cancer (NCT00014547)	Phase II Phase II
Aroplatin	N.A.	Treatment of: Pancreatic neoplasms (NCT00081549)	Phase I/II
Nedaplatin	DNA damage and apoptosis and oxidative stress induction	Treatment of: Nasopharyngeal carcinoma (NCT04834206)	Phase II
NKP1339/Sodium *trans*-tetrachloridobis(1H-indazole)ruthenate(III)	Disturbance of the cellular redox balance, G2/M cell cycle arrest, blockage of DNA synthesis, and induction of apoptosis via the mitochondrial pathway	Treatment of: Solid tumor (NCT01415297)	Phase I
TLD-1433/[Ru(4,4′-dmb)2(IP-3T)]Cl_2_ (4,4′-dmb = 4,4′-dimethyl-2,2′-bipyridine); IP = imidazo[4,5-f][1,10]phenanthroline]; 3T = α-terthienyl	Ideal photophysical properties to act as photodynamic therapy (PDT)	Treatment of: Non-muscle invasive bladder cancer (NMIBC) (NCT03053635)	Phase I
Ferroquine	Negatively regulates Akt kinase and hypoxia-inducible factor-1α (HIF-1α) and redox mechanism	Treatment of: Plasmodium falciparum infection (NCT03660839) Malaria (NCT00563914, NCT05911828)	Phase II Phase I/II
GC-4419/Avasopasem manganese	Superoxide dismutase’s catalytic site as a radiation therapy intervention	Treatment of: Renal impairment (NCT05412472) Squamous cell carcinoma of the oral cavity (NCT01921426)	Phase II Phase II
Rostaporfin Purlytin/Ethyl etiopurpurin (SnET2)	PDT kills cancer cells catalytically in the presence of oxygen and PS produces reactive oxygen species (ROS) and causes cancerous tissues to apoptosis	Treatment of: HIV infections (NCT00002167)	Phase II/III
Aurothiomalate	Inhibit T cells and CD4+ T activation	Treatment of: Non-small-cell lung carcinoma (NSCLC) (NCT00575393)	Phase I
Auranofin/TetraO-acetyl-b-D-(glucopyranosyl)-thio-triethylphosphine	Thioredoxin reductase inhibition (TrxR), which is a crucial enzyme for maintaining redox homeostasis, managing ROS levels, and preventing DNA damage	Treatment of: Amoebic dysentery (NCT02089048) Chronic lymphocytic leukemia (CLL) (NCT01419691) Giardiasis (NCT02736968) Recurrent fallopian tube cancer, Recurrent ovarian epithelial cancer, Recurrent primary peritoneal cavity cancer (NCT01747798) Lung carcinoma (NCT01737502) HIV (NCT02961829)	Phase I Phase II Phase II Phase II Phase I/Phase II Phase II
Cu-ATSM	Due to its low molecular weight, high membrane permeability, and low redox potential, Cu (II)-ATSM easily penetrates cells	Treatment of: Amyotrophic lateral sclerosis (NCT02870634, NCT03136809) Parkinson’s disease (NCT03204929) Diagnosis of: Cervical cancer (NCT00794339)	Phase I, Phase II, Phase I Phase II
Tetrathiomolybdate	Binds copper ions	Treatment of: Wilson disease (NCT00004339) Primary biliary cirrhosis (NCT00805805) Idiophatic pulmunary fibrosis (NCT00189176) Prostate cancer (NCT00150995) Non-small lung cancer (NCT01837329) Esophageal carcinoma (NCT00176800) Psoriasis (NCT00113542) Breast cancer (NCT00195091)	Phase III Phase III Phase I/Phase II Phase II Phase I Phase II Phase II Phase II
Cu^64^-DOTA-alendronate	Detection of small calcium deposits	Diagnosis of: Breast carcinoma (calcification) (NCT03542695)	Early Phase I
Gallium Ga^68^-labeled GRPR Antagonist BAY86-7548	Detection of regional nodal and distant metastases in patients with intermediate- or high-risk prostate cancer	Diagnosis of: Prostate adenocarcinoma (NCT03113617)	Phase II
Sn-117m-DTPA	Targeting of bones affected by metastases	Treatment of: Metastatic prostate adenocarcinoma (NCT04616547)	Phase II

## Data Availability

Not applicable.

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
