# Peer review of "From the Discovery of Targets to Delivery Systems: How to Decipher and Improve the Metallodrugs’ Actions at a Molecular Level"

_pharmaceutics, 2023, doi:10.3390/pharmaceutics15071997_

Round 1

Reviewer 1 Report

1. Cisplatin is discovered by Michele Peyrone (Platinum Metals Rev., 2010, 54, (4), 250). Barnett Rosemberg contributed to an unexpected discovery on its activity (Nature, 205 (1965), pp. 698-699). Please discuss in accurate and cite the references.

2. Please summarize and discuss the clinical stages of metal drugs.

3. Please explain why delivery systems and pro-drug strategies are needed in detail. 

4. Please summarize the targets for the metal drugs introduced here.

5. Please provide a more nuanced view for future development.

Author Response

  1. Cisplatin is discovered by Michele Peyrone (Platinum Metals Rev., 2010, 54, (4), 250). Barnett Rosemberg contributed to an unexpected discovery on its activity (Nature, 205 (1965), pp. 698-699). Please discuss in accurate and cite the references.

We thank the reviewer for his/her comment. We have added the reference about the discovery and synthesis of cisplatin by Michele Peyrone, listed in the revised manuscript as ref. 13, and we modify the main text by adding the following sentence: “The first representative of the chemotherapeutic agents, namely cisplatin, was discovered and synthesized by Michele Peyrone (1813 - 1883) [13], and, its cytotoxic activity was revealed”.

  1. Please summarize and discuss the clinical stages of metal drugs.

We thank the reviewer for his/her suggestion. If we understand the reviewer's request correctly, we need to add a brief summary of the clinical phases. We have added a summary in the introduction section that focuses on the “The Drug development process” established by the FDA Association. We chose to focus on this process ( Drug discovery and development) because it best fits the topics and objective of our review.

  1. Please explain why delivery systems and pro-drug strategies are needed in detail.

We thank the reviewer for this suggestion. We added in section 3. “ Methods to enhance metallodrugs efficiency administration” of the revised manuscript, a paragraph regarding why the use of prodrugs or delivery systems is often crucial for the administration of metallodrugs.

  1. Please summarize the targets for the metal drugs introduced here.

We thank the reviewer for this suggestion. We have added the targets identified in each example reported in the review.

  1. Please provide a more nuanced view for future development.

We thank the reviewer for this suggestion. We modified the future development part to make it soften.

Reviewer 2 Report

The manuscript entitled From the discovery of targets to delivery systems: how to decipher and improve the metallodrugs actions at molecular level focuses on anticancer possibilities and late evolution of metallodrugs, mainly those of Pt, Pd and Ru and some quick screening methods for their fast evaluation.

To my opinion the topic is interesting and well presented, while the manuscript also provides relevant bibliography and informative figures. Nevertheless, prior to publication there are several issues that must be addressed.

Regarding overall presentation, the graphical abstract suggests that there are going to be found many metals in the review (Au, Bi, Mn, Fe, etc.). However, when reading the manuscript, those are not reviewed and some of them are directly not even cited. I would appreciate a clearer graphical abstract in which omitted metals (Bi, Fe, Cu, Zn) do not suggest something that is not going to be covered.

Despite the manuscript could be followed without problems, I would also recommend a thorough language editing, as there are many typos and mistakes even for a non-native reader.

Some figures (specially 1 and 2) are difficult to read due to the small font size and quality image. Please fix this issue.

Additionally, please rewrite the last paragraph on page 5 as it seems to indicate that pulldown technique is also suitable for live cells.

In the second section of the manuscript (Section 3) I would appreciate the following corrected:

·         For better readability, in subsection 1 (hydrolysis), when dealing with the mechanistic aspects of cisplatin, I would indicate Pt oxidation numbers as superindexes.

·         Subsection 2 seems to have missing references not included in bibliography (50 to 54).

·         Subsection 3, photoactivation, has some super/subindexing issues with singlet MLCT and dioxygen.

·         In delivery section, when calling figure 4d, path d could not be clearly identified in such figure.

·         Place adequately references 81 and 82 (both are misplaced).

Names of organic components must be written according to IUPAC: “N” in “N-heterocyclic” and “N, N-dimethyl” must be written in italics. So, the “1H” before benzimidazolyl must be also written in italics.

To conclude, I would also appreciate some scientific aspects more deeply discussed. For example, pulldown measurement strategy is fast and adequate as long as metallodrugs can interact targets. However, the significant number of false negatives and positives pointed herein suggests that inadequate grafting of substrates onto beads disrupts recognition processes. For this reason, I would appreciate a brief discussion on the most convenient grafting strategies developed so far for pharmacophore immobilization.

As commented above, I could follow all explanations along the manuscript. However, there are typos, grammar issues and misleading sentences that must be corrected; reason for which I recommend language editing.

Author Response

The manuscript entitled From the discovery of targets to delivery systems: how to decipher and improve the metallodrugs actions at molecular level focuses on anticancer possibilities and late evolution of metallodrugs, mainly those of Pt, Pd and Ru and some quick screening methods for their fast evaluation.

To my opinion the topic is interesting and well presented, while the manuscript also provides relevant bibliography and informative figures. Nevertheless, prior to publication there are several issues that must be addressed.

We wanted to thank the reviewer for appreciating our work and for all the suggestions given.

Q1. Regarding overall presentation, the graphical abstract suggests that there are going to be found many metals in the review (Au, Bi, Mn, Fe, etc.). However, when reading the manuscript, those are not reviewed and some of them are directly not even cited. I would appreciate a clearer graphical abstract in which omitted metals (Bi, Fe, Cu, Zn) do not suggest something that is not going to be covered. 

Regarding the graphical abstract, we removed the metals that are not deeply discussed throughout the paper.

Q2. Despite the manuscript could be followed without problems, I would also recommend a thorough language editing, as there are many typos and mistakes even for a non-native reader. 

We thank the reviewer for this suggestion. We are sorry about that, we made language editing throughout the paper.

Q3. Some figures (specially 1 and 2) are difficult to read due to the small font size and quality image. Please fix this issue.

We thank the reviewer for his/her suggestion. We increased the fonts in Figures 1 and 2. 

Q4. Additionally, please rewrite the last paragraph on page 5 as it seems to indicate that pulldown technique is also suitable for live cells. 

We thank the reviewer for his/her comment. We modified the sentence underlying that we did not intend the pulldown as a strategy working in living cells.

Q5. In the second section of the manuscript (Section 3) I would appreciate the following corrected:

  •         For better readability, in subsection 1 (hydrolysis), when dealing with the mechanistic aspects of cisplatin, I would indicate Pt oxidation numbers as superindexes. 
  •         Subsection 2 seems to have missing references not included in bibliography (50 to 54).
  •         Subsection 3, photoactivation, has some super/subindexing issues with singlet MLCT and dioxygen. 
  •         In delivery section, when calling figure 4d, path d could not be clearly identified in such figure. 
  •         Place adequately references 81 and 82 (both are misplaced).

We thank the reviewer for these comments, we have made all highlighted corrections.

Q6. Names of organic components must be written according to IUPAC: “N” in “N-heterocyclic” and “N, N-dimethyl” must be written in italics. So, the “1H” before benzimidazolyl must be also written in italics. 

We thank the reviewer for his/her suggestion and have made the suggested corrections.

Q7. To conclude, I would also appreciate some scientific aspects more deeply discussed. For example, pulldown measurement strategy is fast and adequate as long as metallodrugs can interact targets. However, the significant number of false negatives and positives pointed herein suggests that inadequate grafting of substrates onto beads disrupts recognition processes. For this reason, I would appreciate a brief discussion on the most convenient grafting strategies developed so far for pharmacophore immobilization. 

We thank the reviewer for the suggestion. We included an additional discussion about pharmacophore immobilization (page 6). We have also better clarified that the presence of false negatives and positives are not only due to the inadequate grafting of pharmacophores on the beads, but they are general drawbacks in affinity purification-mass spectrometry (AP-MS) strategies employing different kinds of baits to fish the partners (e.g. proteins, organic drugs etc). 

Comments on the Quality of English Language

As commented above, I could follow all explanations along the manuscript. However, there are typos, grammar issues and misleading sentences that must be corrected; reason for which I recommend language editing.

We thank the reviewer for the suggestion. We extensively made language editing throughout the manuscript.

Reviewer 3 Report

Reviewers’ comments for the Manuscript ID:

The manuscript title: “From the discovery of targets to delivery systems: how to decipher and improve the metallodrugs actions at molecular level”

In this mini-review, authors briefly described the about different purification and identification strategies of protein targets of metal compounds, which include affinity purification approaches, including labeling and cross-linking (photo-affinity) strategies for the identification of transient interactions. Furthermore, in the last part of the review authors also describe the different methodologies used nowadays to improve the efficiency of metallodrugs in terms of activation such hydrolysis, redox activation, photoactivation, and delivery system using nanoparticle formation or encapsulate into liposomes. It is well organized manuscripts could be suitable for publication in “Pharmaceutics.

Comments

1)      In the introduction part below sentence is incomplete “including carboplatin and oxaliplatin, have been developed to improve therapeutic efficacy and reduce systemic toxicity [14],[15],[16]. including carboplatin and oxaliplatin, have been developed to improve therapeutic efficacy and reduce systemic toxicity [14],[15],[16]. Systemic toxicity of what?

2)      In page 2 , followed by As compounds, need to correct.

I) Too general English

Author Response

The manuscript title: “From the discovery of targets to delivery systems: how to decipher and improve the metallodrugs actions at molecular level”

In this mini-review, authors briefly described the about different purification and identification strategies of protein targets of metal compounds, which include affinity purification approaches, including labeling and cross-linking (photo-affinity) strategies for the identification of transient interactions. Furthermore, in the last part of the review authors also describe the different methodologies used nowadays to improve the efficiency of metallodrugs in terms of activation such hydrolysis, redox activation, photoactivation, and delivery system using nanoparticle formation or encapsulate into liposomes. It is well organized manuscripts could be suitable for publication in “Pharmaceutics.

We wanted to thank the reviewer for appreciating our work and considering it suitable for publication.

Comments

1)      In the introduction part below sentence is incomplete “including carboplatin and oxaliplatin, have been developed to improve therapeutic efficacy and reduce systemic toxicity [14],[15],[16]. including carboplatin and oxaliplatin, have been developed to improve therapeutic efficacy and reduce systemic toxicity [14],[15],[16]. Systemic toxicity of what?

We thank the reviewer for his/her suggestion, we added “caused by cisplatin” to complete the sentence.

2)      In page 2 , followed by As compounds, need to correct.

We thank the reviewer for his/her suggestion. If we understood the reviewer's comment correctly, we have to refer to As explicitly as arsenic, for this reason we decided to change the sentence as follows: “followed by arsenic (As)”.

Round 2

Reviewer 1 Report

1. For the question 2, the authors are encouraged to discuss the clinical development of metal drugs. It is better to have a table to show the metal drugs in clinical trials. This is the most important aspect of drug development.

2. When discussing the delivery systems and pro-drug strategies, drug resistance may be a key point deserving to be discussed in detail. This paper may be useful (10.1016/j.jconrel.2022.03.049).

Author Response

  1. For the question 2, the authors are encouraged to discuss the clinical development of metal drugs. It is better to have a table to show the metal drugs in clinical trials. This is the most important aspect of drug development.

We thank the reviewer for his/her suggestion. We added in the Introduction part of the main text the following paragraph describing the clinical phases for a metallodrug: 

“Specifically, the clinical trial begins with Phase 0, a preliminary study in which a few healthy subjects are studied at an administration dose of less than 1% of the therapeutic dose for a maximum of seven days, followed by Phase I. After these phases, the metal drug can be tested in affected patients, which describes the phase II of clinical trials. The efficacy of the drug has been tested in patients and a placebo is used as a control. The final step is then the phase III trials, which account for most of the cost of clinical trials. This step aims to confirm the safety tested in phase I and the efficacy tested in phase II (PMID: 24456146). There are several metallodrugs approved by the United States and/or European Union (EU) countries for the medical treatment of human diseases. The uses of these approved metallodrugs range from “Anticancer Metallodrugs” or “Therapeutic Radiopharmaceuticals or Phototherapeutic Metallodrugs" ” to “Antimicrobial and Antiparasitic Metallodrugs”, but also “Antidiabetes” and others (see ref PMID: 24456146 for more information). Prior to the entire preclinical and clinical investigation, the first step is, of course, the investigation of one or more disease targets and the discovery of the molecular mechanisms of action for the metallodrug to be analyzed. This first step is the one addressed in this review (“discovery and development” step), where the potential of the drug as a therapeutic agent will be explored.” 

In addition, we added a table with some metallodrugs currently in clinical trials for different pathologies according to www.clinicaltrials.gov database.

  1. When discussing the delivery systems and pro-drug strategies, drug resistance may be a key point deserving to be discussed in detail. This paper may be useful (10.1016/j.jconrel.2022.03.049).

A2. We thank the reviewer for this suggestion. To satisfy reviewer’s requests, we added in section 3 a detailed description of the drug resistance focusing on cisplatin and the strategies used to counter it. In particular, we added: “Interestingly, nanoparticle-based drug delivery systems have been shown to play a role in overcoming cancer-related drug resistance. Drug resistance is the tolerance of cancer cells to anticancer drugs and is a well-known phenomenon responsible for the limited chemotherapy success. An example of this is given by cisplatin, whose resistance has been a major clinical impediment. Although the mechanism of cisplatin resistance has not been fully clarified, efforts have mainly focused to decrease its deactivation, increase its intracellular accumulation, and reduce DNA repair. The advent of nanomedicines has provided a breakthrough in this area. One of the most prominent examples is NC-6004 (Nanoplatin), a cisplatin nanoparticle developed using a cutting-edge micelle nanotechnology. Cisplatin is encapsulated into polymeric micelles (30 nm size) through the polymer-metal complex formation between polyethylene glycol poly (glutamic acid) block copolymers (PEG-P(Glu)). NC-6004 utilizes the phenomenon known as enhanced permeability and retention (EPR) effect to selectively penetrate and accumulate within tumor lesions. Unlike conventional cisplatin, NC-6004 can circulate in the bloodstream for an extended period. This extended circulation is made possible by the outer PEG shell that protects the micelle from being retained by the reticuloendothelial systems. As a result, NC-6004 gradually releases cisplatin by exchanging chloride ions in the human body.